# Low Cycle Fatigue Life Prediction Model of 800H Alloy Based on the Total Strain Energy Density Method

**DOI:** 10.3390/ma13010076

**Published:** 2019-12-22

**Authors:** Wei Zhang, Tao Jiang, Liqiang Liu

**Affiliations:** College of Mechanical and Electric Engineering, Changchun University of Science and Technology, Changchun 130022, Jilin Province, China; jiangtao@cust.edu.cn (T.J.); 13843155571@163.com (L.L.)

**Keywords:** 800H alloy, strain control, low-cycle fatigue, total strain energy density method, life prediction model

## Abstract

In this paper, a high-temperature low-cycle fatigue life prediction model, based on the total strain energy density method, was established. Considering the influence of the Masing and non-Masing behavior of materials on life prediction, a new life prediction model was obtained by modifying the existing prediction model. With an 800H alloy of the heat transfer tube of a steam generator as the research object, the high-temperature and low-cycle fatigue test was carried out at two temperatures. The results show that the predicted and experimental results are in good agreement, proving the validity of the life prediction model.

## 1. Introduction

Life prediction is an important process that needs to be considered when using metal materials. It can effectively avoid catastrophic failures in nuclear power plants, pipeline systems, offshore platforms, petrochemicals, the aerospace industry, and other related structures. At present, there are many life prediction methods, including the linear cumulative damage method [1], plastic exhaustion method [2], Coffin–Manson correction method [3], strain range division method [4], strain energy range division method [5], metallography method [6], and the stress relaxation range method [7]. Although scholars have studied and furthered the development of these models, equipment damage due to creep-fatigue damage still persists, indicating that accuracy of the existing life prediction models is not sufficient to meet the life-span analysis conditions of different equipment or materials. In response to this problem, scholars have made corrections to existing models, resulting in newer ones. For example, Rao Jin [8], through transformation of the load spectrum based on the linear cumulative damage theory, proposed a functional relationship between the load-holding time and lifetime. Yongqin Wang [9] proposed a new method to predict average plastic strain life based on the energy method. According to the creep fatigue load, the material generates irreversible micro-defects (dislocations, voids, micro-cracks) with the increase of internal energy. The corresponding energy law formula is listed. Runzi Wang [10,11], based on the linear cumulative damage rule, combined with the time fraction method and the plastic exhaustion method, obtained a new strain energy density depletion model by substituting the strain energy constitutive equation.

The above life prediction models were validated for relevant alloy materials and showed their practicability. Incoloy alloy has excellent high-temperature mechanics, corrosion resistance, and oxidation resistance. It is often used in the petrochemical and nuclear power industries, especially for heat transfer tubes in nuclear power plant steam generators. Periodic loading with high-temperature steam reduces the life of the material. Therefore, it is necessary to study the high-temperature and low-cycle fatigue life of Incoloy alloy. Based on the total strain energy density method, a low-cycle fatigue life prediction model for the 800H alloy was proposed. The rationality of the life prediction model was verified based on the data of low-cycle fatigue testing at 675 °C and 750 °C.

## 2. Experiment Material and Method

(1) Experiment material [12]: The test material was an 800H forged alloy produced by Shanghai Jinshun Stainless Steel Material CO. Ltd (Shanghai, China) and a 16-mm diameter steel bar obtained by multiple hot forging treatments of the slab. The main chemical composition of the 800H alloy is as follows: 0.063 C, 0.81Mn, 0.05 Si, 20.6 Cr, 30.67 Ni, 0.47 Ti, 0.05 Cu, 0.02 N, 0.42 Al, 0.007 P, and 0.005 S.

(2) Experiment method: The 16-mm diameter steel bars were cut into two groups. One set of samples was heat treated, while the other was not. For the materials that were heat treated, the room-temperature sample was first heated to 1050 °C for 100 min, kept for 30 min, and water-cooled with a cooling rate of 2 °C/s for 8 min. Then, a box-type resistance furnace (SX-8-10) was heated to 910 °C, kept for 2 h, and cooled with a furnace so that the rate of cooling was 0.006 °C/s. The heat-treated sample was finally obtained. In order to distinguish the effect of heat treatment, the microstructure was analyzed by a metallographic microscope (Leica DM2700, Beijing Ruike Zhongyi Technology Co., Ltd., Beijing, China). The combined metallographic pictures of the samples with heat treatment and without are shown in Figure 1.

A relatively uniform metallographic structure can be obtained by heat treatment, as shown in Figure 1.

The round bar sample after heat treatment was processed into a test sample following size specifications outlined in [13]. The specific dimensions are shown in Figure 2. Considering the impact of cutting methods on the mechanical properties of materials [14], D. Martínez Krahmer [15] studied if an abrasive waterjet, a wire electrodischarge machining, a laser, and punching could be used, instead of the standardised milling process, to obtain tensile specimens, taking into consideration the integrity of surfaces on the specimen. Based on the results of the study mentioned above, A. Suárez [16] proposed a new cutting method—ultrasound vibration assisted milling. This method can effectively improve the surface integrity of the final samples and reduce the effects on the fatigue performance of the material. Thus, a test sample of the 800H alloy was made by using an ultrasound vibration assisted milling lathe and a grinding machine.

Two sets of samples with heat treatment and without were subjected to the low-cycle fatigue test using a high-temperature electronic universal testing machine (Sinotest Equipment Co., Ltd., Changchun, China). Since the occurrence of creep is related to the melting point temperature of the alloy, in this study, it occurred when the test temperature was greater than 0.5*T_m_*. Incoloy 800H has a melting point temperature of 1350 °C to 1400 °C. In order to achieve creep, a creep fatigue test is performed at test temperatures of 675 °C and 750 °C. During the test, the strain control method is adopted; the strain amplitude is 0.3% and 0.02%, the strain ratio R = 0.8, and the strain rates are 5 × 10^−5^/*s*^−1^ and 3.3 × 10^−5^/*s*^−1^. The symmetric cyclic loading waveform of the strain control is shown in Figure 3.

## 3. Modified Total Strain Energy Density Method

In the low-cycle fatigue test of strain control in the cyclic loading process, plastic deformation is the main factor of energy consumption, and thus the calculation of plastic strain energy should first consider the total energy calculation in the process. The material has Masing characteristics, which will determine the calculation method.

### 3.1. Life Prediction Model

(1) Masing Behavior

The most common way to determine the Masing behavior of materials is to plot the hysteresis loop of the stress range and strain range at half-life with different strain amplitudes and the same strain ratio, as shown in Figure 4. The so-called Masing behavior places the minimum stress point of different strain amplitude half-life hysteresis loops at the origin. If the hysteresis loop coincides at half a week and the maximum stress point is on the same line, this confirms that the material has Masing behavior; if the opposite is true, the material has non-Masing behavior.

In this paper, the main consideration is the effect of heat treatment and ambient temperature on the life. There are only two kinds of strain amplitude. It is impossible to determine whether the material has Masing behavior with hysteresis loops of different strain amplitudes, and the error is large. Another method is used, wherein the area of the hysteresis loop is compared with the calculated value of the plastic strain energy constitutive equation to determine whether the material exhibits Masing behavior.

According to Figure 5a, the plastic strain energy equation with the hysteresis loop area of the Masing behavior material is first proposed by A.A. Griffith [17].
(1)$$\Delta W^{p}=\frac{1-n^{\prime }}{1+n^{\prime }}\Delta \sigma \Delta \varepsilon^{p}$$
where $\Delta \varepsilon^{p}$—Plastic strain range; $n^{\prime }$—Cyclic strain hardening index,log(Δσ/2) and $\log(\Delta \varepsilon_{p}/2)$—curve slope; Δσ—stress range.

According to Figure 5b, S.R. Varanasi [18] determined that the calculation formula for plastic strain energy of the non-Masing material is:(2)$$\Delta W_{area1}=\Delta W_{area2}+\Delta W_{area3}$$

This formula is mainly used to consider the influence of stress increment $\delta \sigma_{0}$, thus:(3)$$\Delta W_{area2}=\frac{1-n^{\prime }}{1+n^{\prime }}(\Delta \sigma -\delta_{0}\sigma )\Delta \varepsilon^{p}$$
(4)$$\Delta W_{area2}=\int_{0}^{\Delta \varepsilon^{p}}\Delta \sigma^{*}d\varepsilon^{p}$$
(5)$$\Delta W_{area1}=\int_{0}^{\Delta \varepsilon^{p}}\sigma^{*}d\varepsilon^{p}+\delta \sigma_{0}\Delta \varepsilon^{p}$$

Substituting Equations (4) and (5) into Equation (2):(6)$$\Delta W_{area3}=\delta \sigma_{0}\Delta \varepsilon^{p}$$

Furthermore, the plastic strain energy of non-Masing material is obtained:(7)$$\Delta W^{p}=\Delta W_{area1}=\frac{1-n^{\prime }}{1+n^{\prime }}(\Delta \sigma -\delta \sigma_{0})\Delta \varepsilon^{p}+\delta \sigma_{0}\Delta \varepsilon^{p}$$

After simplification,
(8)$$\Delta W^{p}=\frac{1-n^{\prime }}{1+n^{\prime }}\Delta \sigma \Delta \varepsilon^{p}+\frac{2n^{\prime }}{1+n^{\prime }}\delta \sigma_{0}\Delta \varepsilon^{p}$$
where stress increment $\delta \sigma_{0}$ is
(9)$$\delta \sigma_{0}=\Delta \sigma -\Delta \sigma^{*}=\Delta \sigma -2K^{\prime }{\left(\frac{\Delta \varepsilon^{p}}{2}\right)}^{n\prime }$$
where $K^{\prime }$—cycle strength coefficient, the ordinate intercept of curve of log(Δσ/2) and $\log(\Delta \varepsilon_{p}/2)$ when $\Delta \varepsilon_{p}/2=1$.

In fact, in Equation (8), it is not difficult to find that the results of Equation (8) and Equation (1) are the same when the stress increments $\delta \sigma_{0}=0$, indicating that Masing material is calculated in non-Masing material calculation equation, as a special case.

(2) Plastic strain energy method

M.A. Pompetzki [19] considers energy consumption in addition to plastic deformation. In addition, energy dissipation is caused by sliding along the crystal plane and dislocation motion. Therefore, K. Golos [20] and D. Lefebvre [21] proposed the plastic strain energy life prediction model:(10)$$\Delta W^{p}=\kappa {\left(2N_{f}\right)}^{a}+\Delta W_{0}^{p}$$
where $2N_{f}\to \infty$, $\Delta W^{p}\to \Delta W_{0}^{p}$ when the material constant κ>0, a<0; so, when $2N_{f}<5\times 10^{5}$, $\Delta W_{0}^{p}$ can be ignored.

(3) Total strain energy density method

Based on the plastic strain energy method, F. Ellyin [22] solved the equation for total strain energy density by solving the area of OABDO (Figure 5).
(11)$$\Delta W=\frac{1}{2}\Delta W^{p}+\frac{1}{2}\Delta \sigma \Delta \varepsilon$$

Considering the total strain energy density ΔW and $2N_{f}$, there is a power relationship:(12)$$\Delta W=\kappa^{\prime }{\left(2N_{f}\right)}^{a^{\prime }}+\Delta W_{0}^{}$$

When $\kappa^{\prime }>0$, $a^{\prime }<0$; $2N_{f}\to \infty$, $\Delta W^{}\to \Delta W_{0}^{}$ we further determine that
(13)$$\Delta W_{0}^{}=\left(\Delta W_{0}^{p}+\frac{\Delta \sigma^{2}}{2E}\right)\approx \frac{\Delta \sigma^{2}}{2E}$$

(4) Modified total strain energy density method

By summarizing the research results of the above scholars, it is finally determined that total strain energy consumption factors include plastic deformation, elastic deformation, and sliding dislocation motion (induced by stress increment). Combined with Figure 5, the hysteresis loop and parameter calibration diagram to calculate the total strain energy density of the test parameters is plotted in Figure 6. When $\sigma_{\min}>0$, the total strain energy density $\Delta W_{t}$ is the sum of the three areas in the figure, which is:(14)$$\Delta W_{t}=\Delta W_{p}+\Delta W_{e}+\Delta W_{s}$$
where $\Delta W_{e}$—elastic strain energy density, and $\Delta W_{s}$—additional energy consumption density due to stress increment.

By solving the three regions in Figure 6 to determine the parameters in Equation (14), the results are as follows:(15)$$\Delta W_{p}=\frac{1-n^{\prime }}{1+n^{\prime }}\Delta \sigma \Delta \varepsilon_{p}$$
(16)$$\Delta W_{e}=\frac{1}{2}\Delta \sigma \cdot \Delta \varepsilon_{e}$$

Substituting $\Delta \varepsilon_{e}=\frac{\Delta \sigma }{E}$ into Equation (16),
(17)$$\Delta W_{e}=\frac{\Delta \sigma^{2}}{2E}$$
(18)$$\Delta W_{s}=\sigma_{\min}\Delta \varepsilon$$

After substituting Equations (15), (17), and (18) into Equation (14), total strain energy density can be obtained as:(19)$$\Delta W_{t}=\frac{1-n^{\prime }}{1+n^{\prime }}\Delta \sigma \Delta \varepsilon_{p}+\frac{\Delta \sigma^{2}}{2E}+\sigma_{\min}\Delta \varepsilon$$

The above equation is the modified equation of the total strain energy density of a material with Masing behavior. For non-Masing materials, we simply convert plastic strain energy density according to Equation (8). 

The determined modified total strain energy density $\Delta W_{t}$ and $2N_{f}$ satisfies the power relationship according to Equation (12):(20)$$\Delta W_{t}=A{\left(2N_{f}\right)}^{B}+C$$

### 3.2. Life Prediction Result

It is estimated from the experimental data that the material has Masing behavior at an ambient temperature of 675 °C, and the material does not have Masing behavior at an ambient temperature of 750 °C.

(1) Behaviour at 675 °C

Determining the hardening index $n^{\prime }=0.95336$ by the slope of the slope equation shown in Figure 7a, we calculate the total strain energy density component according to Equation (19). The specific results are shown in Table 1. Then, we determine $A=6.38\times 10^{-6}$ and B=1.52268 according to the slope equation shown in Figure 8a, and according to Formula (13), we infer that $C\approx \frac{\Delta \sigma^{2}}{2E}$. Substituting this into Equation (20), we determine the modified total strain energy density life prediction model.
(21)$$\Delta W_{t}=6.38\times 10^{-6}{\left(2N_{f}\right)}^{1.52268}$$

(2) Behaviour at 750 °C

We determine the hardening index $n^{\prime }=-8.70002$, $K^{\prime }\approx 0$ by the slope of the slope equation shown in Figure 7b and combine Equations (8), (9) and (19) into simplifications.
(22)$$\Delta W_{t}=\Delta \sigma \Delta \varepsilon_{p}+\frac{\Delta \sigma^{2}}{2E}+\sigma_{\min}\Delta \varepsilon$$

The specific results are shown in Table 1. Then, we determined A=70 and B=−0.80644, according to the slope equation shown in Figure 8b, according to formula (13), and infer $C\approx \frac{\Delta \sigma^{2}}{2E}$. Substituting this into (20) and (22), we determine the modified total strain energy density life prediction model.
(23)$$\Delta W_{t}=70{\left(2N_{f}\right)}^{-0.80644}$$

Figure 9 is a fitting curve between the prediction model based on the modified total strain energy density method and the measured values. It can be seen from the figure that the measured values are almost uniformly distributed on the upper and lower sides of the fitting curve. When the ambient temperature is 675 °C, the point without heat treatment (WT point) deviates farther, and the point with heat treatment is evenly distributed, indicating that heat treatment has a great influence on the life expectancy of the cycle fatigue test at this temperature. When the ambient temperature is 750 °C, almost all points are evenly distributed on the fitting curve. It is proven that heat treatment has less influence on cycle test results when the ambient temperature is higher than the creep temperature.

Figure 10 shows a linear fitting result of the 95% confidence interval between the predicted lifetime and measured lifetime of the total strain energy density model corrected at 675 °C and 750 °C. The fitting results at both temperatures are good—within the confidence interval—which proves that the method can be applied to life prediction of the 800H alloy.

## 4. Conclusions

(1) The correction model of the total strain energy density of a material with Masing behavior is proposed on the basis of the plastic strain energy method. For materials without Masing behavior, the plastic strain energy can be modified. The method is reasonable and has a certain theoretical basis for practical engineering applications.

(2) The heat treatment effect is obvious under low-medium temperature test conditions, and not apparent under high-temperature conditions.

(3) The above model is used to predict the life of the strain-controlled low-cycle fatigue of the 800H alloy under high temperature. It was found to have good prediction results for life fluctuations at different temperatures, especially for life prediction under high-temperature test conditions.

## Figures and Tables

**Figure 1 materials-13-00076-f001:**
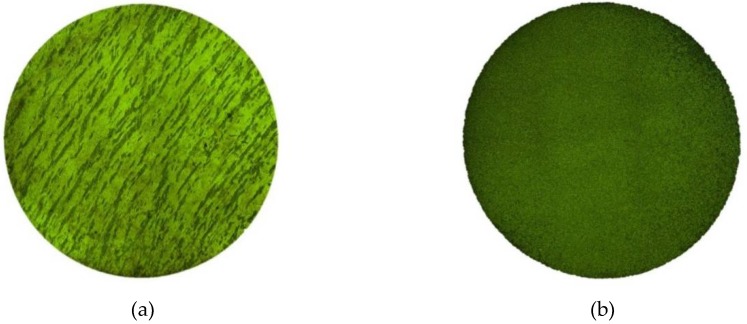
Microstructure of the 800H alloy. (**a**) Sample without heat treatment, (**b**) sample with heat treatment.

**Figure 2 materials-13-00076-f002:**
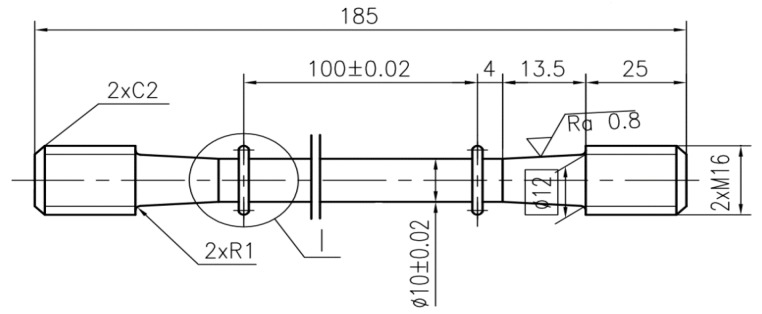
Sample size.

**Figure 3 materials-13-00076-f003:**
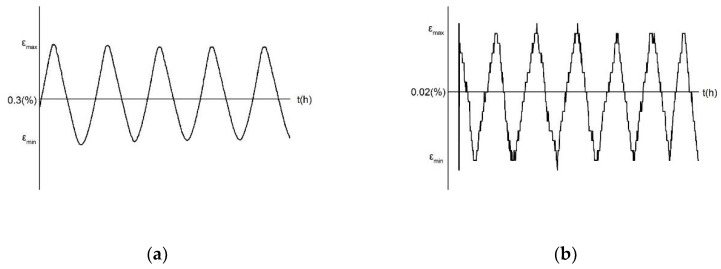
The symmetric cyclic loading waveform based on strain control. (**a**) Test temperature is 675 °C, (**b**) Test temperature is 750 °C.

**Figure 4 materials-13-00076-f004:**
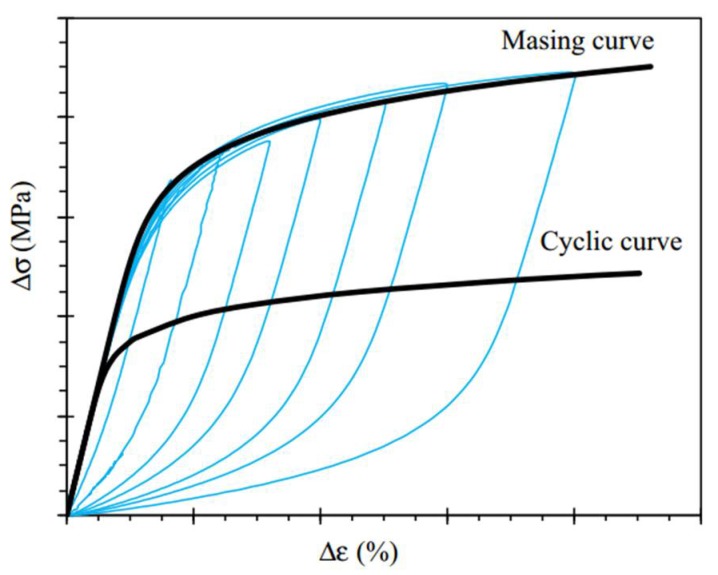
Stress range–strain range hysteresis loop at half-life.

**Figure 5 materials-13-00076-f005:**
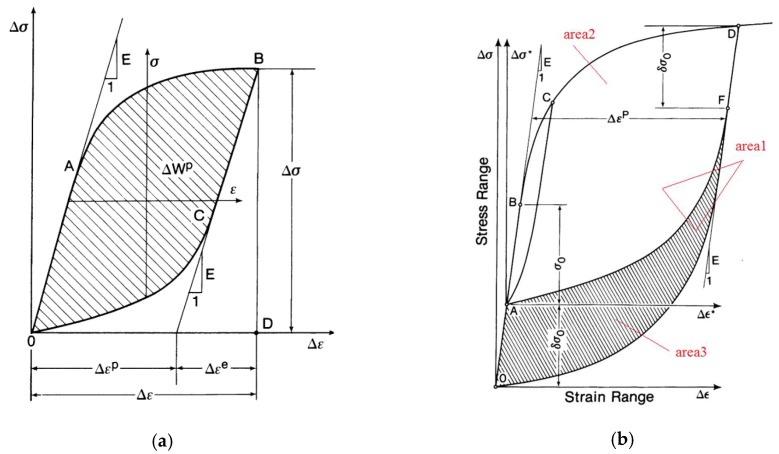
Hysteresis loop at half-life. (**a**) Material with Masing behavior, (**b**) Material with non-Masing behavior.

**Figure 6 materials-13-00076-f006:**
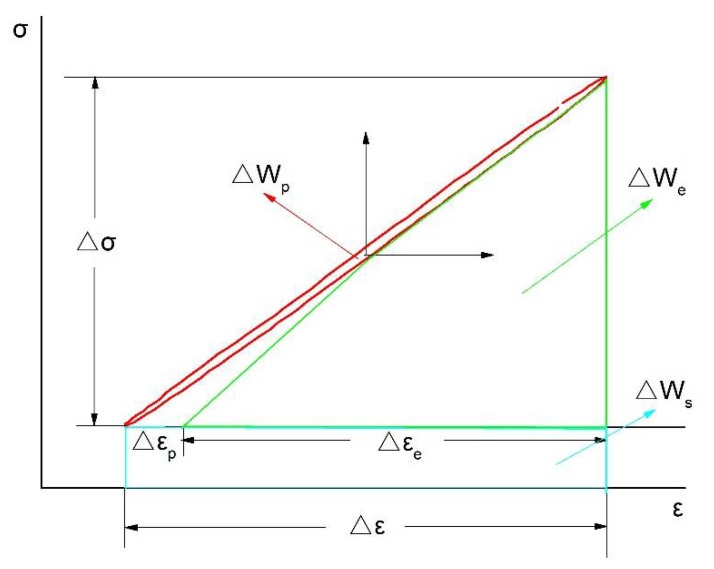
Hysteresis loop and parameter map for energy calculation.

**Figure 7 materials-13-00076-f007:**
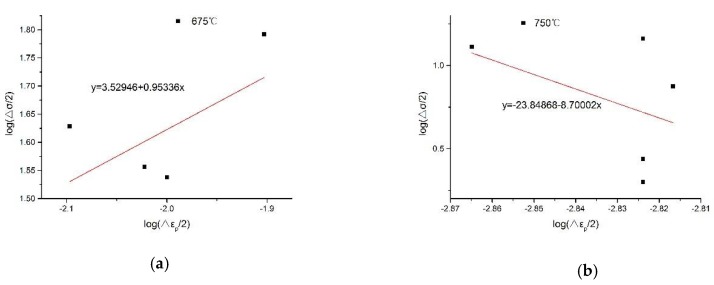
Hardening exponent solution diagram of the slope equation. (**a**) Test temperature is 675 °C, (**b**) Test temperature is 750 °C.

**Figure 8 materials-13-00076-f008:**
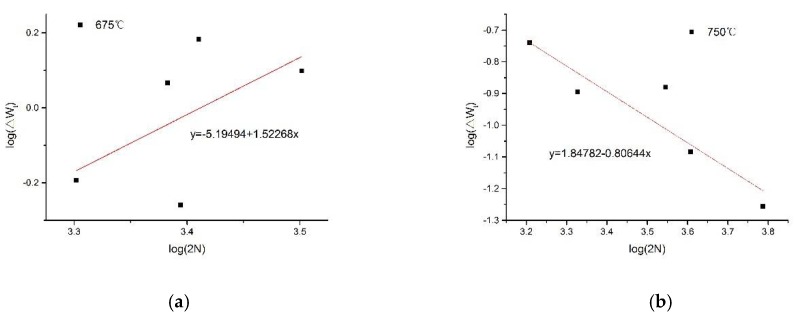
Material constant solution diagram of the slope equation. (**a**) Test temperature is 675 °C, (**b**) Test temperature is 750 °C.

**Figure 9 materials-13-00076-f009:**
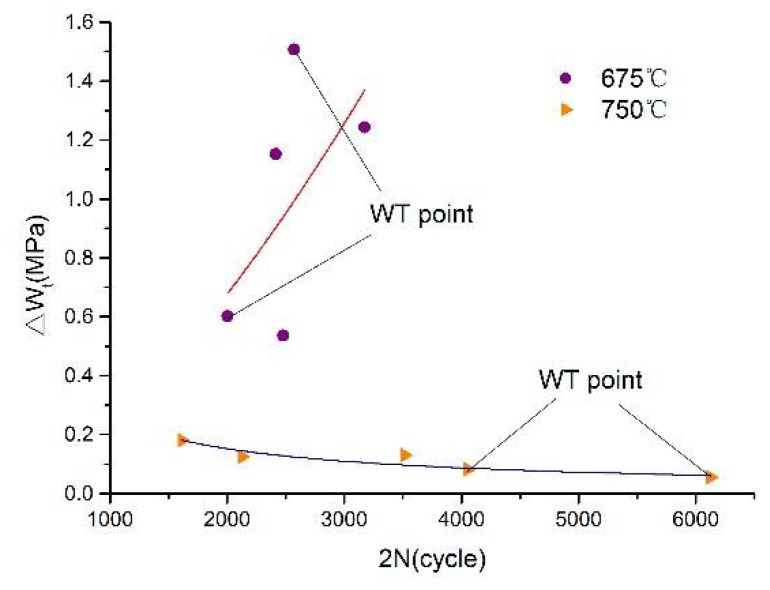
The fitting curve of the modified total strain energy density method.

**Figure 10 materials-13-00076-f010:**
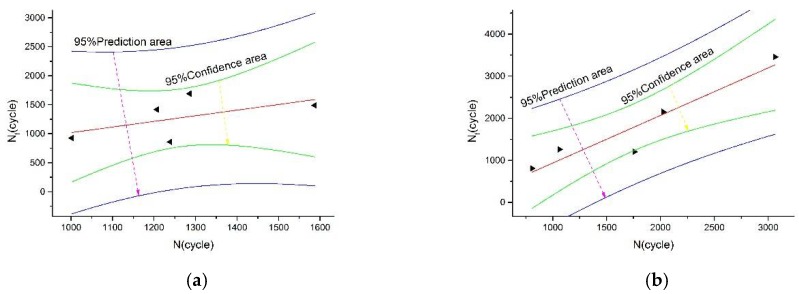
The linear fit graph for life expectancy and measured life. (**a**) Test temperature is 675 °C, (**b**) Test temperature is 750 °C.

**Table 1 materials-13-00076-t001:** Calculation results of the corrected total strain energy of the density method.

Temperature (°C)	Strain Amplitude (%)	No.	Heat treatment	$\Delta W_{p}\;(\mathrm{MPa})$	$\Delta W_{e}\;(\mathrm{MPa})$	$\Delta W_{s}\;(\mathrm{MPa})$	$\Delta W_{t}\;(\mathrm{MPa})$	N	$N_{f}$
675	0.3	1	×	0.0744	0.03912	0.528	0.6415	1002	925
		2	√	0.0331	0.01211	1.21	1.2552	1587	1489
		3	√	0.0328	0.01319	1.12	1.1660	1207	1417
		4	√	0.0328	0.01319	0.504	0.5500	1239	858
		5	×	0.0326	0.01838	1.475	1.5260	1286	1690
750	0.02	6	√	0.0458	0.00057	0.0854	0.1317	1757	1203
		7	√	0.0870	0.00214	0.093	0.1821	807	808
		8	√	0.0710	0.00172	0.0546	0.1273	1062	1262
		9	×	0.0120	0.00004	0.0435	0.0555	3061	3459
		10	×	0.0165	0.00008	0.066	0.0826	2025	2150

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
