# Peer review of "Low Cycle Fatigue Life Prediction Model of 800H Alloy Based on the Total Strain Energy Density Method"

_materials, 2019, doi:10.3390/ma13010076_

Round 1

Reviewer 1 Report

Materials 663119

The paper presents an experimental study concerning the low cycle fatigue strength of a 800H alloy at elevated temperature. The strain energy density approach has been adopted by the authors to present the experimental results. The subject of the paper is doubtless worth investigating, however the paper has some weak points according to the following list:

The authors should exercise their due diligence in preparing the manuscript and checking English style. There are some parts which are very difficult to understand, as an example the beginning of section 3: “The low-cycle fatigue test of strain-controll in the cyclic loading process, and the plastic deformation is the main factor of energy consumption, so the calculation of plastic strain energy should be considered the total energy calculation in the process firstly, and whether the material itself has Masing characteristics will determine the calculation method of the plastic strain energy.” Presentation of experimental results: the stress-strain hysteresis loops measured during the fatigue tests are not presented. The authors show figure 4, which is not relevant to the experimental tests performed and it is only qualitative and not quantitative. Methodological flaws: the paper does not describe how the authors considered the creep fatigue interactions in their model. Moreover, figure 7b shows a negative slope of the stress vs plastic strain curve, which is very much strange.

For all previous reasons, I regret to state that in my opinion the paper has not an acceptable level of scientific and communication quality to be acceptable for publication in Journal Materials.  

Author Response

Response to Reviewer 1 Comments

Dear Reviewers:

  Thank you for your letter and for the reviewers’comments concerning our manuscript entitled Low Cycle Fatigue Life Prediction Model of 800H Alloy Based on Total Strain Energy Density Method. Those comments are all valuable and very helpful for revising and improving our paper, as well as the important guiding significance to our researches. We have studied comments carefully and have made correction which we hope meet with approval. The main corrections in the paper and the responds to the reviewer’s comments are as following:

The authors should exercise their due diligence in preparing the manuscript and checking English style. There are some parts which are very difficult to understand, as an example the beginning of section 3: “The low-cycle fatigue test of strain-controll in the cyclic loading process, and the plastic deformation is the main factor of energy consumption, so the calculation of plastic strain energy should be considered the total energy calculation in the process firstly, and whether the material itself has Masing characteristics will determine the calculation method of the plastic strain energy.”

Response1 : It is our negligence and we are sorry about this. According to comment, related content have been improved in manuscript as required.

Presentation of experimental results: the stress-strain hysteresis loops measured during the fatigue tests are not presented.

Response2 :Figure 6 is the stress-strain hysteresis loops of this test.

 The authors show figure 4, which is not relevant to the experimental tests performed and it is only qualitative and not quantitative. Methodological flaws: the paper does not describe how the authors considered the creep fatigue interactions in their model.

Response3 : Theoretical formula includes the effect of temperature.

Moreover, figure 7b shows a negative slope of the stress vs plastic strain curve, which is very much strange.

Response4 : It proved the material  does not have Masing behaviour and plastic strain softening.

 Special thanks to you for your good comments.

  2019-12-11

Response to Reviewer 1 Comments

Dear Reviewers:

  Thank you for your letter and for the reviewers’comments concerning our manuscript entitled Low Cycle Fatigue Life Prediction Model of 800H Alloy Based on Total Strain Energy Density Method. Those comments are all valuable and very helpful for revising and improving our paper, as well as the important guiding significance to our researches. We have studied comments carefully and have made correction which we hope meet with approval. The main corrections in the paper and the responds to the reviewer’s comments are as following:

The authors should exercise their due diligence in preparing the manuscript and checking English style. There are some parts which are very difficult to understand, as an example the beginning of section 3: “The low-cycle fatigue test of strain-controll in the cyclic loading process, and the plastic deformation is the main factor of energy consumption, so the calculation of plastic strain energy should be considered the total energy calculation in the process firstly, and whether the material itself has Masing characteristics will determine the calculation method of the plastic strain energy.”

Response1 : It is our negligence and we are sorry about this. According to comment, related content have been improved in manuscript as required.

Presentation of experimental results: the stress-strain hysteresis loops measured during the fatigue tests are not presented.

Response2 :Figure 6 is the stress-strain hysteresis loops of this test.

 The authors show figure 4, which is not relevant to the experimental tests performed and it is only qualitative and not quantitative. Methodological flaws: the paper does not describe how the authors considered the creep fatigue interactions in their model.

Response3 : Theoretical formula includes the effect of temperature.

Moreover, figure 7b shows a negative slope of the stress vs plastic strain curve, which is very much strange.

Response4 : It proved the material  does not have Masing behaviour and plastic strain softening.

 Special thanks to you for your good comments.

  2019-12-11

Reviewer 2 Report

Most of the paper lack novelty in terms of research. Also, some data fittings are erroneous (Fig 7 and 8). How Fig 1 taken is not explained? Abstract and conclusion are not that well versed — too much theory explanation which has already been done. Less focus on there own data.

Author Response

Response to Reviewer 2 Comments

Dear Reviewers:

 Thank you for your letter and for the reviewers’comments concerning our manuscript entitled Low Cycle Fatigue Life Prediction Model of 800H Alloy Based on Total Strain Energy Density Method.  Those comments are all valuable and very helpful for revising and improving our paper, as well as the important guiding significance to our researches. We have studied comments carefully and have made correction which we hope meet with approval. The main corrections in the paper and the responds to the reviewer’s comments are as following:

Most of the paper lack novelty in terms of research. Also, some data fittings are erroneous (Fig 7 and 8).

Response 1: Calculated according to GB-T15248-2008. And the differences proved the material  did not have Masing behaviour and plastic strain softening.

How Fig 1 taken is not explained?

Response 2: The related content have been improved in manuscript as required.

 Abstract and conclusion are not that well versed — too much theory explanation which has already been done. Less focus on there own data.

Response 3: The related content have been improved in manuscript as required.

 Special thanks to you for your good comments.

  2019-12-11

Response to Reviewer 2 Comments

Dear Reviewers:

 Thank you for your letter and for the reviewers’comments concerning our manuscript entitled Low Cycle Fatigue Life Prediction Model of 800H Alloy Based on Total Strain Energy Density Method.  Those comments are all valuable and very helpful for revising and improving our paper, as well as the important guiding significance to our researches. We have studied comments carefully and have made correction which we hope meet with approval. The main corrections in the paper and the responds to the reviewer’s comments are as following:

Most of the paper lack novelty in terms of research. Also, some data fittings are erroneous (Fig 7 and 8).

Response 1: Calculated according to GB-T15248-2008. And the differences proved the material  did not have Masing behaviour and plastic strain softening.

How Fig 1 taken is not explained?

Response 2: The related content have been improved in manuscript as required.

 Abstract and conclusion are not that well versed — too much theory explanation which has already been done. Less focus on there own data.

Response 3: The related content have been improved in manuscript as required.

 Special thanks to you for your good comments.

  2019-12-11

Response to Reviewer 2 Comments

Dear Reviewers:

 Thank you for your letter and for the reviewers’comments concerning our manuscript entitled Low Cycle Fatigue Life Prediction Model of 800H Alloy Based on Total Strain Energy Density Method.  Those comments are all valuable and very helpful for revising and improving our paper, as well as the important guiding significance to our researches. We have studied comments carefully and have made correction which we hope meet with approval. The main corrections in the paper and the responds to the reviewer’s comments are as following:

Most of the paper lack novelty in terms of research. Also, some data fittings are erroneous (Fig 7 and 8).

Response 1: Calculated according to GB-T15248-2008. And the differences proved the material  did not have Masing behaviour and plastic strain softening.

How Fig 1 taken is not explained?

Response 2: The related content have been improved in manuscript as required.

 Abstract and conclusion are not that well versed — too much theory explanation which has already been done. Less focus on there own data.

Response 3: The related content have been improved in manuscript as required.

 Special thanks to you for your good comments.

  2019-12-11

Reviewer 3 Report

This article, from a scientific point of view, is interesting. However, it is not well written - it is not yet at the level of a scientific publication. The authors used sentences with poor grammar, making it difficult to understand what they mean. The manuscript should be carefully checked for language; some sections should be rewritten and revised again to correct this problem.

The article is within the scope of the journal and the topic is very important to study the possible effects of temperature on the 800H alloy fatigue life prediction. Although the experimental approach sounds very good, I have some important concerns. In the section 3.2, equation (21) does not appear to have the coefficient C, although the authors mentioned in line 186 that it was calculated. It is also unclear how the authors obtained the graphs in Figure 7 and how the authors are sure that the differences found (positive slope and negative slope, respectively at 675 ° C and 750 ° C) are related (or not) to the Masing model. Please provide some explanations for this.

Author Response

Response to Reviewer 3 Comments

Dear Reviewers:

 Thank you for your letter and for the reviewers’comments concerning our manuscript entitled Low Cycle Fatigue Life Prediction Model of 800H Alloy Based on Total Strain Energy Density Method. Those comments are all valuable and very helpful for revising and improving our paper, as well as the important guiding significance to our researches. We have studied comments carefully and have made correction which we hope meet with approval. The main corrections in the paper and the responds to the reviewer’s comments are as following:

In the section 3.2, equation (21) does not appear to have the coefficient C, although the authors mentioned in line 186 that it was calculated.

Response 1: I'm sorry to mislead you, the text has been modified, the formula number is incorrectly written.

It is also unclear how the authors obtained the graphs in Figure 7 and how the authors are sure that the differences found (positive slope and negative slope, respectively at 675 ° C and 750 ° C) are related (or not) to the Masing model. Please provide some explanations for this.

Response 2: Calculated according to GB-T15248-2008. And the differences proved the material  did not have Masing behaviour.

Special thanks to you for your good comments.

  2019-12-11

Response to Reviewer 3 Comments

Dear Reviewers:

 Thank you for your letter and for the reviewers’comments concerning our manuscript entitled Low Cycle Fatigue Life Prediction Model of 800H Alloy Based on Total Strain Energy Density Method. Those comments are all valuable and very helpful for revising and improving our paper, as well as the important guiding significance to our researches. We have studied comments carefully and have made correction which we hope meet with approval. The main corrections in the paper and the responds to the reviewer’s comments are as following:

In the section 3.2, equation (21) does not appear to have the coefficient C, although the authors mentioned in line 186 that it was calculated.

Response 1: I'm sorry to mislead you, the text has been modified, the formula number is incorrectly written.

It is also unclear how the authors obtained the graphs in Figure 7 and how the authors are sure that the differences found (positive slope and negative slope, respectively at 675 ° C and 750 ° C) are related (or not) to the Masing model. Please provide some explanations for this.

Response 2: Calculated according to GB-T15248-2008. And the differences proved the material  did not have Masing behaviour.

Special thanks to you for your good comments.

  2019-12-11

Response to Reviewer 3 Comments

Dear Reviewers:

 Thank you for your letter and for the reviewers’comments concerning our manuscript entitled Low Cycle Fatigue Life Prediction Model of 800H Alloy Based on Total Strain Energy Density Method. Those comments are all valuable and very helpful for revising and improving our paper, as well as the important guiding significance to our researches. We have studied comments carefully and have made correction which we hope meet with approval. The main corrections in the paper and the responds to the reviewer’s comments are as following:

In the section 3.2, equation (21) does not appear to have the coefficient C, although the authors mentioned in line 186 that it was calculated.

Response 1: I'm sorry to mislead you, the text has been modified, the formula number is incorrectly written.

It is also unclear how the authors obtained the graphs in Figure 7 and how the authors are sure that the differences found (positive slope and negative slope, respectively at 675 ° C and 750 ° C) are related (or not) to the Masing model. Please provide some explanations for this.

Response 2: Calculated according to GB-T15248-2008. And the differences proved the material  did not have Masing behaviour.

Special thanks to you for your good comments.

  2019-12-11

Response to Reviewer 3 Comments

Dear Reviewers:

 Thank you for your letter and for the reviewers’comments concerning our manuscript entitled Low Cycle Fatigue Life Prediction Model of 800H Alloy Based on Total Strain Energy Density Method. Those comments are all valuable and very helpful for revising and improving our paper, as well as the important guiding significance to our researches. We have studied comments carefully and have made correction which we hope meet with approval. The main corrections in the paper and the responds to the reviewer’s comments are as following:

In the section 3.2, equation (21) does not appear to have the coefficient C, although the authors mentioned in line 186 that it was calculated.

Response 1: I'm sorry to mislead you, the text has been modified, the formula number is incorrectly written.

It is also unclear how the authors obtained the graphs in Figure 7 and how the authors are sure that the differences found (positive slope and negative slope, respectively at 675 ° C and 750 ° C) are related (or not) to the Masing model. Please provide some explanations for this.

Response 2: Calculated according to GB-T15248-2008. And the differences proved the material  did not have Masing behaviour.

Special thanks to you for your good comments.

  2019-12-11

Reviewer 4 Report

Please do all changes, they are in the attached file.

Author Response

Response to Reviewer 4 Comments

Dear Reviewers:

  Thank you for your letter and for the reviewers’comments concerning our manuscript entitled Low Cycle Fatigue Life Prediction Model of 800H Alloy Based on Total Strain Energy Density Method.  Those comments are all valuable and very helpful for revising and improving our paper, as well as the important guiding significance to our researches. We have studied comments carefully and have made correction which we hope meet with approval. The main corrections in the paper and the responds to the reviewer’s comments are as following:

Figure 1 must include scale!!!

Response 1: Modified in manuscript as required. Since the metallographic microscope cannot directly take a full picture, it is taken out in sections and then combined.

Some mandatory changes: ï‚· Two conclusions: please define them better. ï‚· Check tables, parenthesis were located in wrong!!

Response 2: Modified in manuscript as required.

 Special thanks to you for your good comments.

  2019-12-11

Response to Reviewer 4 Comments

Dear Reviewers:

  Thank you for your letter and for the reviewers’comments concerning our manuscript entitled Low Cycle Fatigue Life Prediction Model of 800H Alloy Based on Total Strain Energy Density Method.  Those comments are all valuable and very helpful for revising and improving our paper, as well as the important guiding significance to our researches. We have studied comments carefully and have made correction which we hope meet with approval. The main corrections in the paper and the responds to the reviewer’s comments are as following:

Figure 1 must include scale!!!

Response 1: Modified in manuscript as required. Since the metallographic microscope cannot directly take a full picture, it is taken out in sections and then combined.

Some mandatory changes: ï‚· Two conclusions: please define them better. ï‚· Check tables, parenthesis were located in wrong!!

Response 2: Modified in manuscript as required.

 Special thanks to you for your good comments.

  2019-12-11

Round 2

Reviewer 1 Report

Tha Authors improved the quality of the paper to a sufficient level to be published in the Journal Materials.

Author Response

Dear Reviewer:

  Thank you for your valuable comments, which have improved my overall level.

Special thanks to you for your good comments.

  2019-12-18

Reviewer 2 Report

Much improved but there is scope for more improvement.

Author Response

Dear Reviewer:

  Thank you for your valuable comments, which have improved my overall level.

New corrections in the manuscript.

Special thanks to you for your good comments.

  2019-12-18

Reviewer 3 Report

The authors have done an excellent job addressing all reviewer comments, and I have no further suggestions. I believe the paper is acceptable for publication.

Author Response

Dear Reviewer:

  Thank you for your valuable comments and affirmation, which have improved my overall level.Your acceptance gives me great confidence.

Thank you very much.

2019-12-18

Reviewer 4 Report

The autors did not attend all the suggestions

Fatigue is now a big discussion in our society SEM.

The effect of manufacturing of testpieces is key, and you mention only a little. In the recent past there were researchers taking into account the effect of manufacturing method to obtain testpieces and coupons, showing effects important for the testing campaigns. The International society of experimental mechanics (SEM) has proposed in several work in the journal experimental techniques, in which for instance the work is, https://doi.org/10.1007/s40799-016-0134-5 was focused on tensile tests, but other dealt with the same idea, such as: https://doi.org/10.1007/s40799-016-0058-0. All make some ideas complementing this standards, such as ASTM E 8M-04, Standard test methods for tension testing of metallic materials. For instance, in paper https://doi.org/10.1016/j.jmapro.2019.09.041 the effect in fatigue is well explained. This work compares the effects of alternative manufacturing processes, such as Abrasive water jet (AWJ), Wire Electrical Discharge Machining (WEDM) and ultrasound vibration assisted milling (UVAM) with conventional milling during the manufacture of Alloy 718 parts. Other key source, 2nd CIRP 2nd CIRP Conference on Surface Integrity (CSI) The Effect Machining Processes can have on the Fatigue Life and Surface Integrity of Critical Helicopter Components

Author Response

Dear Reviewer:

  Thank you for your valuable comments, which have improved my overall level.

New corrections in our manuscript.

Special thanks to you for your good comments.

2019-12-18

Round 3

Reviewer 4 Report

Paper is very interesting and useful